# Patient-Factors Influencing the 2-Year Trajectory of Mental and Physical Health in Prostate Cancer Patients

Alessandro Cicchetti [1], Marianna Noale [2,*], Paola Dordoni [1], Barbara Noris Chiorda [1], Letizia De Luca [1], Lara Bellardita [1], Rodolfo Montironi [3], Filippo Bertoni [4], Pierfrancesco Bassi [5], Riccardo Schiavina [6], Mauro Gacci [7], Sergio Serni [7], Francesco Sessa [7], Marco Maruzzo [8], Stefania Maggi [2], Riccardo Valdagni [1,9,10,†] on behalf of The Pros-IT CNR Study Group

1   Prostate Cancer Program, Fondazione IRCCS Istituto Nazionale dei Tumori, 20133 Milan, Italy
2   Aging Branch, Neuroscience Institute, National Research Council, 35128 Padua, Italy
3   Molecular Medicine and Cell Therapy Foundation c/o, Polytechnic University of the Marche Region, 60121 Ancona, Italy
4   Prostate Group, Italian Association for Radiation Oncology (AIRO), 20124 Milan, Italy
5   Department of Urology, Policlinico Gemelli, Catholic University of Rome, 00168 Rome, Italy
6   Division of Urology, IRCCS Azienda Ospedaliero-Universitaria di Bologna, 40138 Bologna, Italy
7   Department of Urologic Robotic Surgery and Renal Transplantation, Careggi Hospital, University of Florence, 50134 Florence, Italy
8   Medical Oncology Unit 1, Veneto Institute of Oncology IOV-IRCCS, 35128 Padua, Italy
9   Department of Oncology and Hemato-Oncology, Università degli Studi di Milano, 20122 Milan, Italy
10  Unit of Radiation Oncology, Fondazione IRCCS Istituto Nazionale dei Tumori, 20133 Milan, Italy
*   Correspondence: marianna.noale@in.cnr.it; Tel.: +39-0498218899
†   The Pros-IT CNR study group members are listed at the end of the article.

**Abstract:** This study aimed to examine the physical and mental Quality of Life (QoL) trajectories in prostate cancer (PCa) patients participating in the Pros-IT CNR study. QoL was assessed using the Physical (PCS) and Mental Component Score (MCS) of Short-Form Health Survey upon diagnosis and two years later. Growth mixture models were applied on 1158 patients and 3 trajectories over time were identified for MCS: 75% of patients had constantly high scores, 13% had permanently low scores and 12% starting with low scores had a recovery; the predictors that differentiated the trajectories were age, comorbidities, a family history of PCa, and the bowel, urinary and sexual functional scores at diagnosis. In the physical domain, 2 trajectories were defined: 85% of patients had constantly high scores, while 15% started with low scores and had a further slight decrease. Two years after diagnosis, the psychological and physical status was moderately compromised in more than 10% of PCa patients. For mental health, the trajectory analysis suggested that following the compromised patients at diagnosis until treatment could allow identification of those more vulnerable, for which a level 2 intervention with support from a non-oncology team supervised by a clinical psychologist could be of help.

**Keywords:** prostate cancer; health related quality of life; SF-12; growth mixture model

## 1. Introduction

Prostate cancer (PCa) is one of the leading cancers diagnosed in adult males worldwide, with an estimated incidence of 1,436,000 new cases and an age-standardized incidence rate of 49.9 per 100,000 person-years [1]. Even if the stage of cancer detection with an early diagnosis is important for most cancer survival, for PCa, considering the extremely high one- and five-year survival rates, the stage of detection could be less important [2]. Cutting-edge diagnostic tools and treatments have improved numerous patients' quality of life (QoL). However, it is well established that PCa patients may show early signs of psychological distress (e.g., anxiety, depression) in addition to physical problems [3,4]. PCa diagnosis represents a stressful life event that, together with treatment, can significantly impact the

patient's psycho-emotional status and QoL. Although medium to long-term physical and mental QoL trajectories seem to differ and to be relatively stable in many patients [4], depression, anxiety, signs of post-traumatic stress disorder, pain, sexual problems, difficulty in urinating, along with other disturbances or symptoms, have frequently been reported during the initial and later stages of the disease [5]. Indeed, for non-metastatic PCa patients, anxiety and depression appear to be at their highest levels during the pre-treatment phase. Men reported significantly less anxiety, better mental health and feeling of depression following the initial phase of the treatment [6,7], with treatment decision-making having an impact on patients QoL [8].As the psychological well-being of PCa patients is critically important, and adjustment to the disease is positively related to QoL levels [9–11], more knowledge about the possible trajectories and evolution of both physical and psychological states in patients facing PCa is needed. However, when evaluating longitudinal data, the heterogeneity in QoL trajectories among patients within a population may be masked by analyses based on mean effects; the growth mixture models (GMM) approach could be interesting since it assesses the existence of different trajectories within a population when grouping variables are not known a priori [12,13]. For this reason, the current work described the physical and mental QoL trajectories in Italian male adults diagnosed with non-metastatic PCa to determine who could benefit from personalised care support tools enabling the best possible clinical and personal outcomes [14]. Patients participating in the Pros-IT CNR study and monitored over two years after diagnosis were considered and data examined using the GMMs approach; socio-demographic and clinical variables were analysed with treatment patterns as potential predictors of the trajectories.

## 2. Materials and Methods

### 2.1. Participants

The design of the Pros-IT CNR project has been described in detail elsewhere [15]. It is a longitudinal, observational study aiming to monitor QoL in a sample of Italian patients diagnosed with biopsy-verified prostate cancer, beginning in September 2014. Ninety-seven Urology, Radiation Oncology and Medical Oncology facilities in Italy were involved in the project, and 1705 treatment-naïve patients were enrolled. A baseline assessment at the time of diagnosis and evaluations 6, 12, 24, 36, 48, and 60 months later were foreseen by protocol [16]. The data collected during the baseline assessment included demographic and anamnestic information, the initially formulated diagnosis, the cancer stage, the risk factors, comorbidities, and health-related QoL scores. Data regarding the cancer treatments and patients' QoL scores were collected at each follow-up examination.

The Ethics Committee of the coordinating centre (Sant'Anna Hospital, Como, Italy; register number 45/2014) and all the hospitals or health care facilities involved in the project approved the study protocol. The study was carried out according to the Declaration of Helsinki principles, and all of the participants signed informed consent forms.

### 2.2. Outcome Variables

The patients' general QoL was assessed using the Italian version of the Short-Form Health Survey (SF-12 Standard v1 scale) [17], which is composed of two summary measures: the Physical Component Score (PCS) and the Mental Component Score (MCS). The SF-12 is a widely recognised, reliable, and valid measure of health-related QoL commonly used in multicenter trials. Indeed, Gandek et al. showed that "for large group comparisons and longitudinal monitoring, the differences in measurement reliability of the SF-12 and SF-36 are less important". In fact, in a study such as this one, which focuses on "measuring overall physical and mental health outcomes rather than an eight-scale profile, the SF-12 may be advantageous" [18]. The score on each domain and the total score of each patient were computed using the algorithms suggested by Apolone et al. [17]. The possible range of scores on each section is between 0 and 100, with 100 indicating the best self-perceived health.

*2.3. Predictor Variables*

The patients' socio-demographic variables at diagnosis, their clinical variables, including comorbidities, the Gleason score, the clinical T-score, the prostate-specific antigen (PSA) level at diagnosis, as well as their PCa treatments, and their urinary, bowel, and sexual QoL at the time of diagnosis were considered as predictors.

The PCa treatments carried out up to the 24-month follow-up assessment were classified as follows:

1. Active surveillance (AS). The patients who did not remain in the group up to the 24-month follow-up were excluded;
2. Nerve-sparing radical prostatectomy (NSRP);
3. Non-nerve sparing exclusive radical prostatectomy (NNSRP);
4. Exclusive radiotherapy (RT);
5. Radiotherapy plus androgen deprivation therapy (RT plus ADT, not considering patients on ADT after radiotherapy for cancer recurrence).

Patients treated with adjuvant radiotherapy, adjuvant ADT after prostatectomy, or brachytherapy were not included in our analyses. The same applied to the patients who dropped out of active surveillance (Figure 1).

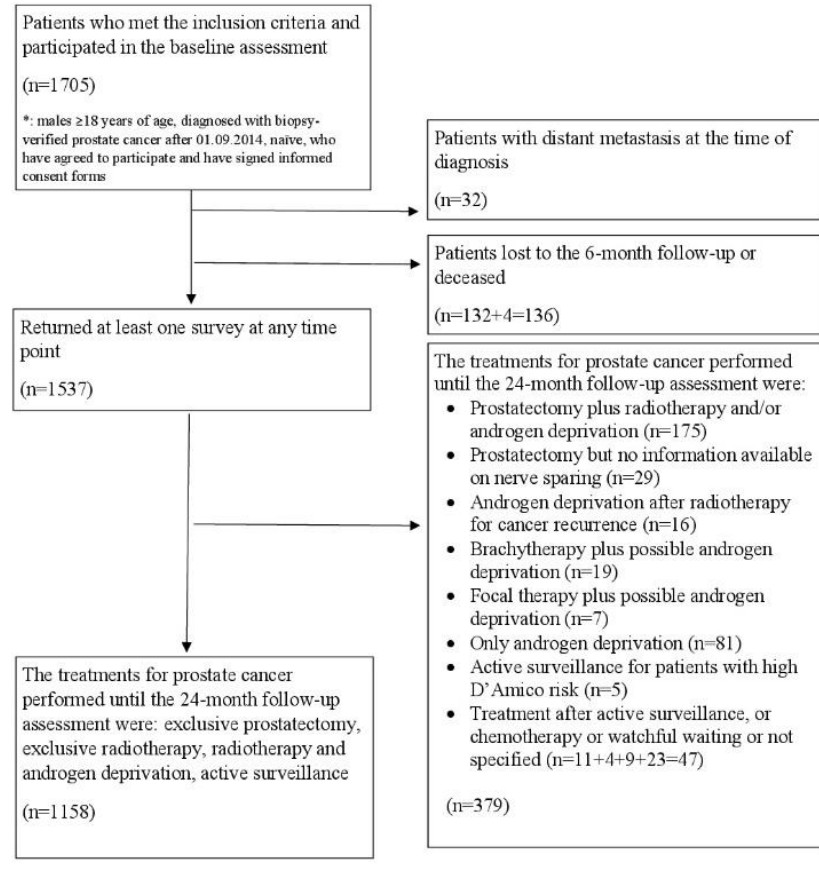

**Figure 1.** Flow Diagram of the Pros-IT patients.

The University of California Los Angeles-Prostate Cancer Index (Italian UCLA-PCI) [19,20] was used to evaluate urinary (UF), bowel (BF) and sexual (SF) function.

### 2.4. External Comparison

The MCS-12 and PCS-12 score distributions of the participants in the Pros-IT CNR project were graphically and statistically compared with those of an extensive survey conducted at the beginning of 2000 by the Italian Institute of Statistics (ISTAT) of the general Italian population and the oncologic group that also used the SF-12 questionnaire. The survey was carried out on a sample of 140,000 citizens; for the comparisons, we selected (i) the group of cancer patients ($n$ = 598) and (ii) the group of male citizens within the same age range as the patients enrolled in Pros-IT ($n$ = 14,291).

Raw data from that investigation were not available; statistical information regarding the distribution of the PCS and the MCS values were retrieved from the literature [17]. Mean values and the standard deviation were used to generate the data distribution.

### 2.5. Statistical Analysis

The data of the participants who underwent one or more of the follow-up assessments were analysed; missing values were not imputed. Summary statistics are expressed as means ± standard deviation (SD) or median and (Quartile 1 (Q1), Quartile 3 (Q3)) for quantitative variables and frequency percentages for categorical variables.

GMMs were applied to identify trajectories within the Pros-IT CNR population. When individuals are expected to experience different changes over time both in terms of strength and direction, a simple mean based trajectory could mask differences; modelling techniques considering heterogeneity in change over time could be preferred [21], and GMMs are statistical techniques that have been used to describe group differences in changes over time, estimating an average growth curve for each identified class, calculating intercept, slope and growth parameter variance by maximizing the log-likelihood function [12]. For each individual, the probability of belonging to each identified class is estimated on the bases of observed data, and participants are assigned to the group with the higher posterior probability, considering also the possible contribution of covariates [22]. SAS Proc Traj was used to identify the subgroups of participants with similar QoL trajectories [23,24], and the following steps were considered:

- the optimal number of groups was identified by fitting several models ranging from single to 5-group models;
- the shape of the trajectories was identified considering polynomials of varying degrees for each group, starting with a cubic specification, and then dropping non-significant polynomial terms;
- The model fit statistics (Bayesian Information Criterion (BIC)), the value of group membership probability and the average posterior probability (entropy) were considered to identify the best model:
  - ○ the magnitude of the difference in the BIC (2ΔBIC > 10) was used to choose between less or more complex models.
  - ○ the analysis aimed to identify groups including at least 5% of the population;
  - ○ the average posterior probability of membership was ascertained for each group; values greater than 0.7 indicate adequate internal reliability.

Chi-square or Kruskal-Wallis tests were then applied to evaluate unadjusted differences between the trajectory groups identified. Multinomial logistic regression models were used to evaluate the variables associated with the trajectory groups. Age at diagnosis, education, marital status, living arrangements, family history of prostate cancer, comorbidities, diabetes, body mass index (BMI), the PSA at diagnosis, the Gleason score, the clinical T-Stage, UF, BF, and SF at diagnosis (highest quartile vs lower quartile according to the distribution in the sample), and the PCa treatments were considered independent variables. Models were also adjusted for the time between the end of PCa treatment and the last follow-up assessment the patient underwent.

Additional GMM and logistic models were defined stratifying age according to its median value (<70 vs. $\geq$70 years).

A *t*-test was performed to compare the MCS-12 and PCS-12 scores with the ISTAT population.

All the statistical analyses were performed using SAS software version 9.4 (SAS Institute, Cary, NC, USA).

## 3. Results

Patients (*n* = 1705) were enrolled in the Pros-IT CNR study, and their characteristics at diagnosis have been described in detail elsewhere [16]. Data regarding the PCa treatments the patients underwent, excluding participants with distant metastasis at diagnosis (*n* = 32), were available for 1158 patients and included NSRP (*n* = 311), NNSRP (*n* = 187), RT (*n* = 334), RT plus ADT (*n* = 252) and AS (*n* = 74) (Figure 1).

Patients undergoing other treatments (*n* = 379 patients) were not considered in the present report. One thousand thirty-three participants (89%) underwent the 12-month follow-up assessment, and 804 (69%) underwent the 24-month one. Table 1 presents the characteristics at diagnosis of the patients included in the analyses; patients treated with RT and RT plus ADT were older, had more comorbidities and had higher-risk disease features in comparison with those treated with NSRP, NNSRP or AS.

Regarding the following analysis, we differentiated between:

- a statistically significant difference ($p < 0.05$);
- a minimal clinically important difference (MCID) in the mental or physical domain, i.e., how much of a difference in scores would result in some change in clinical management that is to be considered clinically meaningful [25]. Empirical findings from distribution based methods studies showed a tendency to converge to the $\frac{1}{2}$ SD criteria as a meaningful moderate difference [26,27]. In the following analysis, we considered the conservative estimate approach by Sloan and colleagues for a minimum clinical important difference (MCID = 1 SD) from the patient's perspectives [28,29]. This large effect size considers differences that overcome the limitations due to any subjective (the patient) and objective (the questionnaire) bias or error.

**Table 1.** Characteristics at the time of diagnosis of the Pros-IT population considered.

| | Overall (*n* = 1158) | Nerve-Sparing Exclusive Prostatectomy (*n* = 311) | Non-Nerve-Sparing Exclusive Prostatectomy (*n* = 187) | Exclusive Radiotherapy (*n* = 334) | Radiotherapy and Androgen Deprivation (*n* = 252) | Active Surveillance (*n* = 74) | *p*-Value [§] |
|---|---|---|---|---|---|---|---|
| Age at diagnosis, years, mean $\pm$ SD | 68.8 $\pm$ 7.4 | 63.2 $\pm$ 6.8 | 66.9 $\pm$ 6.1 | 72.8 $\pm$ 5.2 | 72.5 $\pm$ 5.9 | 66.9 $\pm$ 6.5 | <0.0001 |
| Education > lower secondary school, *n* (%) | 562 (49.2) | 178 (57.6) | 100 (53.8) | 144 (43.6) | 97 (39.8) | 43 (58.1) | <0.0001 |
| BMI $\geq$ 30 kg/m$^2$, *n* (%) | 177 (15.6) | 34 (11.1) | 29 (15.5) | 56 (17.1) | 50 (21.0) | 8 (10.8) | 0.0179 |
| Current smoker, *n* (%) | 166 (14.6) | 48 (15.8) | 35 (18.9) | 43 (13.2) | 29 (11.7) | 11 (15.3) | 0.2554 |
| Diabetes mellitus, *n* (%) | 172 (14.9) | 23 (7.4) | 28 (15.0) | 57 (17.2) | 59 (23.4) | 5 (6.8) | <0.0001 |
| 3 + moderate/severe comorbidities *, *n* (%) | 174 (15.0) | 32 (10.3) | 22 (11.8) | 59 (17.7) | 50 (19.9) | 11 (14.9) | 0.0089 |
| Family history of prostate cancer, *n* (%) | 187 (16.3) | 71 (23.1) | 32 (17.5) | 39 (11.7) | 37 (15.0) | 8 (10.8) | 0.0015 |
| T staging at diagnosis, *n* (%) | | | | | | | |
| T1 | 557 (50.2) | 200 (65.6) | 97 (55.4) | 131 (41.6) | 63 (25.9) | 66 (93.0) | <0.0001 |
| T2 | 445 (40.1) | 102 (33.4) | 72 (41.2) | 150 (47.6) | 116 (47.8) | 5 (7.0) | |
| T3 or T4 | 107 (9.7) | 3 (1.0) | 6 (3.4) | 34 (10.8) | 64 (26.3) | 0 (0.0) | |

**Table 1.** *Cont.*

| | Overall (*n* = 1158) | Nerve-Sparing Exclusive Prostatectomy (*n* = 311) | Non-Nerve-Sparing Exclusive Prostatectomy (*n* = 187) | Exclusive Radiotherapy (*n* = 334) | Radiotherapy and Androgen Deprivation (*n* = 252) | Active Surveillance (*n* = 74) | *p*-Value [§] |
|---|---|---|---|---|---|---|---|
| Gleason score at diagnosis, *n* (%) | | | | | | | |
| ≤6 | 535 (46.6) | 186 (60.0) | 76 (40.9) | 155 (47.1) | 48 (19.1) | 70 (98.6) | |
| 3 + 4 | 279 (24.3) | 78 (25.2) | 49 (26.3) | 86 (26.1) | 65 (25.9) | 1 (1.4) | <0.0001 |
| 4 + 3 | 157 (13.7) | 27 (8.7) | 36 (19.4) | 47 (14.3) | 46 (18.3) | 1 (1.4) | |
| ≥8 | 177 (15.4) | 19 (6.1) | 25 (13.4) | 41 (12.5) | 92 (36.7) | 0 (0.0) | |
| PSA at diagnosis, ng/mL, median (Q1, Q3) | 7 (5.1, 10) | 6.3 (5, 8.7) | 6.9 (5.1, 10) | 7 (5.1, 9.9) | 8.9 (6.3, 14.3) | 6.2 (4.9, 7.7) | <0.0001 |
| D'Amico risk class, *n* (%) | | | | | | | |
| Low | 303 (26.7) | 120 (39.1) | 43 (23.6) | 70 (21.4) | 10 (4.0) | 60 (85.7) | |
| Intermediate | 494 (43.5) | 152 (49.5) | 97 (53.3) | 146 (44.7) | 89 (35.7) | 10 (14.3) | <0.0001 |
| High | 338 (29.8) | 35 (11.4) | 42 (23.1) | 111 (33.9) | 150 (60.3) | 0 (0.0) | |
| UCLA PCI UF, mean ± SD | 93.7 ± 15.1 | 96.5 ± 10.7 | 94.2 ± 15.0 | 91.9 ± 17.1 | 92.4 ± 16.5 | 93.8 ± 15.0 | 0.0006 |
| UCLA PCI UB, mean ± SD | 89.1 ± 22.7 | 92.8 ± 20.0 | 92.3 ± 19.5 | 86.2 ± 24.5 | 84.7 ± 25.8 | 92.5 ± 17.0 | <0.0001 |
| UCLA PCI BF, mean ± SD | 93.6 ± 13.4 | 96.1 ± 9.3 | 94.3 ± 12.9 | 91.7 ± 15.4 | 91.8 ± 15.0 | 94.5 ± 12.6 | 0.0004 |
| UCLA PCI BB, mean ± SD | 93.7 ± 17.6 | 92.3 ± 12.9 | 94.6 ± 16.0 | 92.9 ± 18.4 | 90.4 ± 22.7 | 95.9 ± 14.4 | 0.0100 |
| UCLA PCI SF, mean ± SD | 50.2 ± 31.7 | 66.6 ± 27.0 | 56.4 ± 29.2 | 37.9 ± 30.3 | 37.9 ± 29.2 | 61.1 ± 30.2 | <0.0001 |
| UCLA PCI SB, mean ± SD | 63.9 ± 34.8 | 71.8 ± 32.2 | 61.7 ± 35.1 | 58.7 ± 36.5 | 58.8 ± 34.8 | 75.7 ± 27.2 | <0.0001 |
| SF-12 PCS, mean ± SD | 51.9 ± 7.2 | 53.7 ± 5.7 | 52.6 ± 6.7 | 50.8 ± 7.8 | 50.2 ± 8.3 | 52.7 ± 6.1 | <0.0001 |
| SF-12 MCS, mean ± SD | 49.5 ± 9.7 | 49.3 ± 9.4 | 47.9 ± 10.0 | 50.2 ± 9.7 | 49.2 ± 9.9 | 50.9 ± 9.2 | 0.0300 |

SD: Standard Deviation; BMI: Body Mass Index; Q1: Quartile 1; Q3: Quartile 3. SF-12: Short-Form Health Survey; PCS: Physical Component Subscale; MCS: Mental Component Subscale. UCLA: University of California Los Angeles-Prostate Cancer Index; UF: Urinary Function; UB: Urinary Bother; BF: Bowel Function; BB: Bowel Bother; SF: Sexual Function; SB: Sexual Bother. Scores ranges from 0 to 100, with higher scores representing better quality of life in relation to functions or symptoms. * Based on Cumulative Illness Rating Scale (CIRS); [§] *p*-value from Chi-square or Fisher exact tests for categorical variables, generalised linear model after testing for homoschedasticity (Levene test) or Kruskal-Wallis test for continuous variables.

### 3.1. MCS Analysis

#### 3.1.1. MCS at Diagnosis

At diagnosis, the mean MCS baseline value for the whole population was 49.3 with a SD of 9.4, which is the MCID for the mental status (Table S1).

While the average value in the patients undergoing AS (50.9 ± 9.2) was significantly higher than that of those undergoing NNSRP (47.9 ± 10), it was not clinically relevant. This finding will be examined at greater length in the discussion. Since no significant differences were found between AS vs. NSRP, RT or RT plus ADT (49.3 ± 9.4, 50.2 ± 9.7, 49.2 ± 9.9, respectively), the patients had similar mental statuses at the onset before they underwent different cancer treatments. In terms of age, if we consider the median age of our population as a threshold, the mean MCS value in the patients < 70 years old was 49.0 ± 9.6 vs. 49.9 ± 9.8 in those aged ≥70 years (*p* = 0.0282).

#### 3.1.2. MCS over Time

The mean MCS values rose during the first 6 months after diagnosis; they also rose during the following 6 months and then fell between the next 12-month and the 24-month follow-up assessments (Figure 2).

Three trajectories for the MCS scores over the 24-month period analysed were identified (Figure 2; Table S2). We report in this section the baseline intercept mean coefficient (BIM), i.e., the baseline mean score according to the trajectory group, the trajectory score at 2-years of follow-up (2yrFU), the posterior group membership probability (GrMemb), i.e., the likelihood for all the patients within the group to be described by that trajectory

- The "reference group" (Trajectory Group 3 (75% of the patients with GrMemb = 0.97)): the patients in this group showed constantly high scores throughout the 24-month follow-up period. BIM was 53.9, 2yrFU was 51.4.

- The "recovering group" (Trajectory Group 2 (12% of the patients with GrMemb = 0.87)): this group of patients started with low scores at diagnosis, then presented higher values at the 6-month follow-up, which they maintained until the end of the assessment. The difference between the baseline mean value for trajectory 2 members (34.3) and the total population mean value (49.3) exceeded the MCID. The mental health improvement exceeded the MCID in the first six-month follow-up, and then trajectory 2 members had normal range of values for the following follow-up time (Figure 2, black line).
- The "permanently low score group" (Trajectory Group 1 (13% of the patients with GrMemb = 0.92)): this group of patients started with low scores. The scores first fell to an even lower level and then surged upwards. BIM was 39.2, the nadir was 34.2, and the 2yrFU was 44.1. The difference with the total population mean value at the baseline exceeded the MCID. In contrast with the group 2 trajectory, the more considerable discrepancy was recorded at 6 months (34.2), where the average value for the population was 51.0.

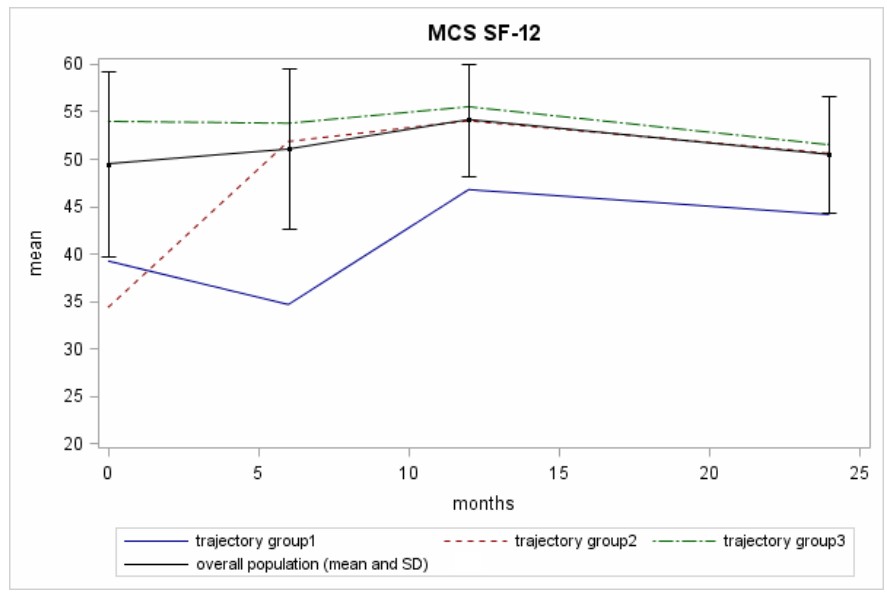

**Figure 2.** The mean MCS scores across the timeline of the 4 evaluations for the Pros-IT participants.

The predictors that significantly differentiated the MCS trajectories, evaluated using multinomial logistic regression analysis and considering Trajectory Group 3 (constantly high group) as a reference, were: age at diagnosis, comorbidities, a family history of prostate cancer, and the UF, BF, and SF scores at diagnosis (Table 2).

With respect to the constantly high group (Trajectory Group 3), patients in the "recovering group" (Trajectory Group 2) were younger at diagnosis and had higher levels of comorbidity. At diagnosis, high comorbidity levels were also significantly associated with Trajectory Group 1 membership ("permanently low score group"). A family history of prostate cancer was also significantly related to both Trajectory 2 and 1 memberships. The UF, BF and SF scores at diagnosis in the lower quartile (i.e., worst self-perceived functions) were associated with Trajectory 3 ("reference group") with respect to Trajectory 2 ("recovering group") membership. The lower quartile of UF and BF score was also associated with Trajectory 3 with respect to Trajectory 1 ("permanently low score group") membership.

A sub-analysis of the three MCS trajectories of the patients older and younger than 70 is included in the Supplementary Materials.

**Table 2.** Predictors of the trajectory class membership for the Mental Composite Score (MCS).

| MCS SF-12 | Trajectory 2 vs. 3 | | Trajectory 1 vs. 3 | |
|---|---|---|---|---|
| | OR (95% CI) | *p*-Value | OR (95% CI) | *p*-Value |
| Age at diagnosis (years) | 0.94 (0.91, 0.97) | 0.0003 | 0.98 (0.95, 1.01) | 0.2278 |
| Education > lower secondary school | 1.13 (0.76, 1.68) | 0.5417 | 1.34 (0.89, 2.00) | 0.1605 |
| Marital status, married vs widowed, divorced or never married | 1.39 (0.51, 3.80) | 0.5257 | 1.12 (0.44–2.82) | 0.8157 |
| Living arrangement, with other vs alone | 1.68 (0.52–5.46) | 0.3908 | 1.84 (0.64–5.28) | 0.2563 |
| BMI $\geq 30$ kg/m$^2$ | 0.82 (0.68, 1.19) | 0.1784 | 1.04 (0.79, 1.38) | 0.9759 |
| Diabetes mellitus | 0.88 (0.61, 1.10) | 0.5628 | 1.00 (0.61, 1.70) | 0.9580 |
| Family history of prostate cancer | 1.87 (1.17, 2.99) | 0.0092 | 1.70 (1.03, 2.82) | 0.0392 |
| 3 + moderate/severe comorbidities * | 1.90 (1.16, 3.11) | 0.0112 | 1.86 (1.15, 3.02) | 0.0114 |
| Current smoker | 0.84 (0.48, 1.46) | 0.5327 | 1.27 (0.74, 2.17) | 0.3865 |
| D'Amico risk class, high vs. intermediate/low | 1.58 (1.00, 2.49) | 0.0501 | 0.95 (0.59, 1.51) | 0.8207 |
| Prostate cancer treatments | | | | |
| NNSRP vs. NSRP | 1.23 (0.70, 2.14) | 0.4757 | 1.13 (0.55, 2.32) | 0.7350 |
| RT vs. NSRP | 0.69 (0.37, 1.27) | 0.2331 | 1.58 (0.82, 3.02) | 0.1716 |
| RT plus ADT vs. NSRP | 0.62 (0.30, 1.28) | 0.1927 | 1.83 (0.88, 3.84) | 0.1082 |
| AS vs. NSRP | 0.86 (0.35, 2.10) | 0.7435 | 1.84 (0.78, 4.35) | 0.1648 |
| Distance between the end of treatment and follow-up assessment, days | 1.01 (0.97, 1.05) | 0.5037 | 1.05 (0.92, 1.10) | 0.7563 |
| UF at diagnosis [§], highest quartile vs. lower [1] | 0.55 (0.35, 0.85) | 0.0075 | 0.52 (0.34,0.79) | 0.0024 |
| BF at diagnosis [§], highest quartile vs. lower [2] | 0.43 (0.29, 0.65) | <0.0001 | 0.36 (0.24, 0.54) | <0.0001 |
| SF at diagnosis [§], highest quartile vs. lower [3] | 0.48 (0.29, 0.80) | 0.0051 | 0.77 (0.45, 1.32) | 0.3369 |

NSRP: Nerve-Sparing Exclusive Radical Prostatectomy; NNSRP: Non Nerve-Sparing Exclusive Radical Prostatectomy; RT: exclusive Radiotherapy; RT plus ADT: Radiotherapy and Androgen Deprivation Therapy; AS: Active Surveillance; UF: Urinary Function; BF: Bowel Function; SF: Sexual Function; * Based on Cumulative Illness Rating Scale (CIRS); [§] Based on University of California Los Angeles—Prostate Cancer Index (UCLA-PCI). [1] UF = 100 vs. <100; [2] BF = 100 vs. <100; [3] SF $\geq$ 80 vs. <80.

*3.2. PCS Analysis*

3.2.1. PCS at Diagnosis

At diagnosis, the mean PCS value was 51.9 with a SD of 7.2 (Table 1). The mean score at diagnosis in the patients undergoing AS (52.7 ± 6.1) was significantly higher than that in the patients undergoing RT or RT plus ADT (50.8 ± 7.8 and 50.2 ± 8.3, respectively; Table 1). There were no significant differences between NNSRP and NSRP (53.7 ± 5.7, 52.6 ± 6.7, respectively).

3.2.2. PCS over Time

Mean PCS values in the overall population were substantially flat over the 24 months analysed (Figure 3).

Two trajectories for the PCS scores in the Pros-IT participants were identified (Figure 3; Table S3):

- The "reference group" (Trajectory Group 2 (85% of the patients with GrMemb = 0.98)): this group of patients showed constantly high scores throughout the 24-month follow-up, with a BIM of 53.2, a 2yrFU of 52.6;
- The "decreasing group" (Trajectory Group 1 (15% of the patients with GrMemb = 0.92)): this group of patients started with low physical scores at diagnosis (BIM = 42.9). The scores fell to an even lower level at the 6-month follow-up, and they continued to decrease until the 24-month follow-up assessment (2yrFU = 37.7). The difference between the baseline mean value for this trajectory group and the overall mean exceeded the MCID. The decline with time increased the distance in PCS for these patients and the trajectory Group 2.

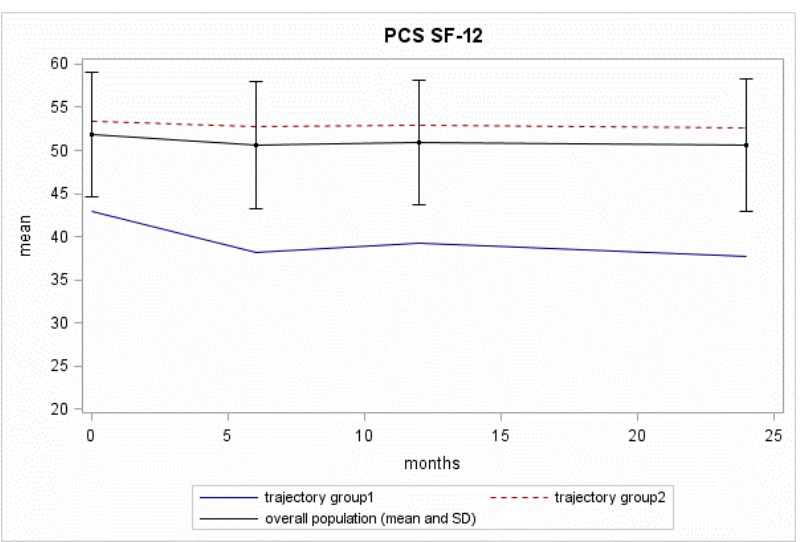

**Figure 3.** The PCS SF-12 scores across the timeline of the 4 evaluations for the Pros-IT participants.

The patient characteristics significantly associated with the PCS trajectories are outlined in Table 3.

**Table 3.** Predictors of the trajectory class membership for the Physical Composite Score (PCS).

| | Class 2 vs. 1 | |
| --- | --- | --- |
| **PCS SF-12** | **OR (95% CI)** | ***p*-Value** |
| Age at diagnosis (years) | 1.02 (0.97, 1.07) | 0.4128 |
| Education > lower secondary school | 0.99 (0.60, 1.65) | 0.9752 |
| Marital status, married vs widowed, divorced or never married | 1.02 (0.32, 3.30) | 0.9691 |
| Living arrangement, with other vs alone | 1.23 (0.33, 4.64) | 0.7632 |
| BMI $\geq$ 30 kg/m$^2$ | 0.97 (0.67, 1.40) | 0.8644 |
| Diabetes mellitus, *n* (%) | 1.99 (1.11, 3.59) | 0.0214 |
| Family history of prostate cancer | 1.02 (0.52, 2.03) | 0.9466 |
| 3 + moderate/severe comorbidities * | 1.23 (0.67, 2.26) | 0.5144 |
| Current smoker, *n* (%) | 1.35 (0.65, 2.82) | 0.4193 |
| D'Amico risk class, high | 0.70 (0.40, 1.23) | 0.2142 |
| Prostate cancer treatments | | |
| NNSRP vs. NSRP | 1.05 (0.35, 3.15) | 0.9327 |
| ER vs. NSRP | 3.01 (1.24, 7.30) | 0.0150 |
| RT plus ADT vs. NSRP | 3.56 (1.18, 10.7) | 0.0246 |
| AS vs. NSRP | 1.19 (0.24, 5.96) | 0.8342 |
| Distance between the end of treatment and follow-up assessment, days | 1.06 (0.98, 1.13) | 0.6156 |
| UF at diagnosis [§], highest quartile vs. lower [1] | 0.55 (0.33, 0.94) | 0.0284 |
| BF at diagnosis [§], highest quartile vs. lower [2] | 0.47 (0.28, 0.78) | 0.0032 |
| SF at diagnosis [§], highest quartile vs. lower [3] | 0.47 (0.21, 1.07) | 0.0727 |

NSRP: Nerve-Sparing Exclusive Radical Prostatectomy; NNSRP: Non Nerve-Sparing Exclusive Radical Prostatectomy; RT: exclusive Radiotherapy; RT plus ADT: Radiotherapy and Androgen Deprivation Therapy; AS: Active Surveillance; UF: Urinary Function; BF: Bowel Function; SF: Sexual Function. * Based on Cumulative Illness Rating Scale (CIRS); [§] Based on University of California Los Angeles—Prostate Cancer Index (UCLA-PCI). [1] UF = 100 vs. <100; [2] BF = 100 vs. <100; [3] SF $\geq$ 80 vs. <80.

The "decreasing group" was associated with high levels of diabetes. Moreover, lower quartile scores at diagnosis for UF and BF were significantly associated with the "decreasing group" membership. A borderline significant protective effect was also found for SF. This finding suggests that patients included in the Trajectory 2 Group had a compromised health condition. RT and RT plus ADT prostate cancer treatments, as opposed to NSRP, were significantly associated with the "decreasing group" trajectory membership.

A sub-analysis of the PCS trajectories younger and older than 70 is included in the Supplementary Materials.

### 3.3. Health Status Comparison with the Italian Population Collected by the ISTAT

The current study compared the PCS and MCS distributions in the Pros-IT study participants with those reported by the ISTAT. Particularly, the PCS and MCS in the men in the same age groups as those in the Pros-IT participants and in the group (male and female) with cancer were analysed. We generated the ISTAT distribution from the mean and standard deviation values included in the report and considered an asymmetrical normal distribution (toward the left tail). The distributions shown are thus not exact and have only a graphical meaning. Tables in Supplementary Materials (Tables S4 and S6) and the *p*-value coming from the *t*-test of the distributions have statistical meaning. Finally, we compared the impact of comorbidities on the PCS and MCS in the two studies.

3.3.1. Age Groups in the Men (Tables S4 and S5)

The ISTAT investigation presented the MCS and PCS in 6 age range divided by gender. We compared the Pros-IT distributions (855 patients) with the values obtained from the ISTAT male participants (14,291 citizens). Figure 4a,c show the distributions and analysis for MCS and PCS, respectively.

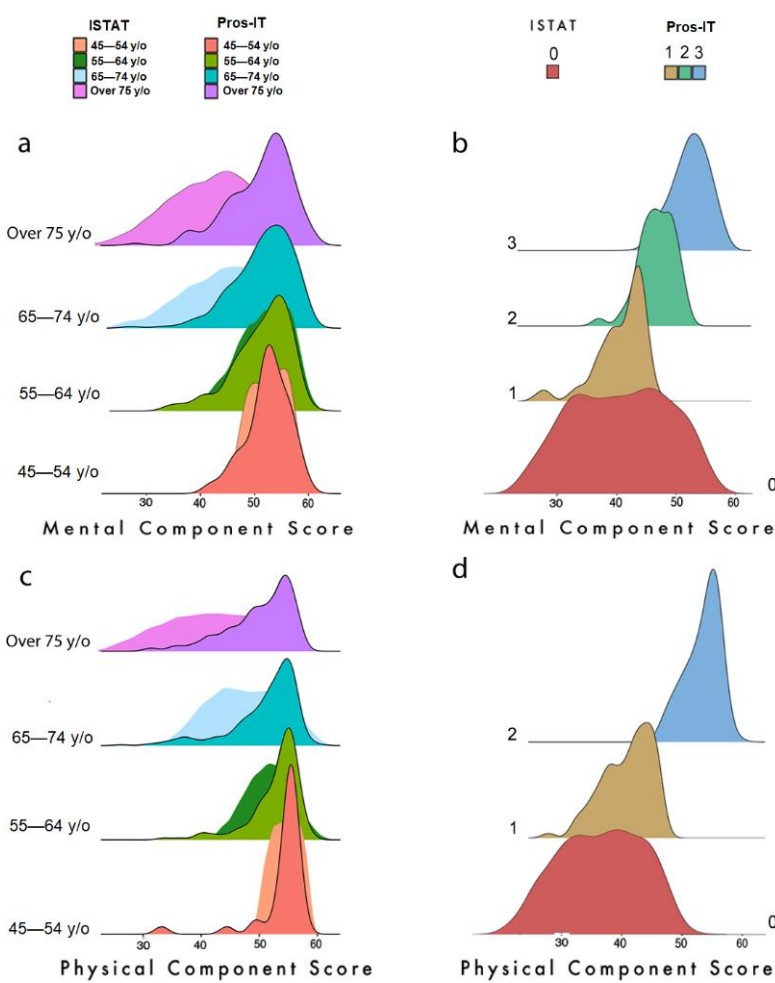

**Figure 4.** (**a**) Distribution of MCS scores of the Pros-IT and ISTAT populations according to age groups; (**b**) Distribution of MCS scores in trajectories for Pros-IT participants and of the ISTAT cancer patients; (**c**) Distribution of PCS scores of the Pros-IT and ISTAT populations according to age groups; (**d**) Distribution of PCS scores in trajectories for Pros-IT participants and of the ISTAT cancer patients.

A *t*-test was used to compare the distributions according to the statistical values reported in Table S4. A significant difference (*p*-value < 0.001) was found for both domains in the 65–74 and 75+ year ranges. For PCS, a significant *p*-value was also found in the 55–64 year-old group.

In both plots Figure 4c,d the ochre distribution depicts the 1st trajectory group; the powder blue one represents the 2nd trajectory group and the green one represents the 3rd trajectory group (defined only with regard to the MCS analysis); the red one represents the scores of the ISTAT population.

### 3.3.2. Cancer Pathology (Tables S6 and S7)

The ISTAT analysis showed the distribution of the mental and physical scales in individuals with several diseases, including cancer pathology. These data were used to compare the global health condition of the prostate cancer patient in Italy (represented by Pros-IT cohort, 684 patients) instead of the state of cancer patients (any cancer, characterised by the ISTAT subgroup of 598 citizens) (Figure 4b,d show the distributions and analysis for MCS and PCS).

To facilitate the comparison, we examined the average distribution overtime for the Pros-IT participants (the average distribution of the MCS/PCS at the five time-points) of the trajectory groups defined by our analysis, i.e., 3 groups for MCS and 2 for PCS.

We can infer from the comparison of the statistics in Table S6 that the mental condition of the Pros-IT patients was superior (*p*-value < 0.001 for the *t*-test comparing the distributions) to that of cancer patients in 87% of the cases. Only patients in Trajectory Group 1 (13% of the Pros-IT population) had a comparable mental state. As far as the PCS was concerned, the Pros-IT prostate cancer patients showed a significantly better average score for both trajectories (*p*-value < 0.001).

### 3.3.3. Impact of Other Diseases on the MCS and PCS (Tables S8 and S9)

Apolone and colleagues showed (reported here in Table S9) the relationship between PCS/MCS classes and the average number of comorbidities. The same table designed for the Pros-IT study (Table S8) confirmed a similar inverse correlation with comorbidities for PCS and MCS. Our findings indicated that the Pros-IT participants in the highest deciles had a mean of less than one comorbidity, while those in the first deciles had a mean of two or more diseases. This was true for both the PCS and MCS.

## 4. Discussion

The current study investigates the physical and mental QoL trajectories in Italian male adults diagnosed with PCa, who have been monitored for more than two years. In general, participants had a good QoL status, although some findings require further consideration.

For the mental domain, a large percentage of patients (trajectory 3) showed good mental health throughout the 24-month follow-up period. At the time of diagnosis, a limited number of patients (trajectory 1 and 2) experienced a clinically meaningful difference from the average value for the Pros-IT cohort. The evolution of these patients showed two different patterns with time. Twelve per cent of patients started with low mental health at the time of diagnosis but recovered 6 months after the event, suggesting that discomfort was likely generated by the cancer event and the uncertainties on the care path for these patients. Indeed, it can be hypothesised that after the patient has come to terms with the diagnosis [30] and has reached an agreement with health care professionals about treatment [10], he perceives less stress and, as a result, shows lower anxiety and better coping strategies (i.e., the constantly changing cognitive and behavioural efforts to manage specific external and/or internal demands) [6,31–33]. Most acute prevalences of depression, anxiety and psychological distress seem to occur before and after the conclusion of treatment, with possible negative impacts on QoL [7].

Another small percentage of patients (13%) showed low mental health at baseline, but they also did not manifest an improvement over time. In fact, in the core phases of

the treatment, they experienced a further decrease in their mental health. Unlike patients in trajectory 2, they could have a limited array of resources to cope with the disease. It is worth noting that half of these patients (around 1/20 patients) were also included in the PCS trajectory 1. Thus, a discrepancy with the average population at two years could be explained by the impact of physical health on mental conditions.

As far as the participants' mental health was concerned, our results showed no clinical differences between the various treatments groups at the time of enrolment. Some predictors of mental health trajectories (younger men with 2 or less moderate/severe comorbidities, no familiarity with PCa, and good mental health at the time of enrolment) were more likely to maintain stable mental health throughout the follow-up period. As previous studies have highlighted, age at diagnosis [34] and comorbidities [35] are critical predicting factors conditioning the path of mental health.

Regarding physical health, the patients with lower scores embarked on their path with a compromised health condition reflected by diabetes mellitus and worse urinary and bowel function. A significant *p*-value was also found for RT and RT plus ADT as opposed to NSRP in Trajectories 1 and 2. It is worth noting that the two trajectories started out with an important difference in the baseline value. Patients treated with surgery had a better health condition (see Table 1 for further details) and were able to face the treatment modality. On the contrary, patients who underwent RT and RT plus ADT worsened with time, but it is not clear if this could be associated with the divergent ways with which the more physically impaired group faced the treatment. The best clinical approach has to be selected for these patients to limit the discrepancy after treatment.

Several studies analysed the longitudinal evolution of physical and mental condition in PCa patients, averaging the scores according to the received treatment. Punnen et al. [36] reported similar trend; in particular, mental health remained stable over time with little difference across treatments while the adjusted physical function had a decline at 2 years, but no differences were highlighted between surgical and non-surgical treatments. Hoofman et al. performed a similar analysis dividing by PCa with favorable and unfavorable risk disease [37]; none of the treatment groups reported a clinically meaningful decline in physical function, emotional well-being, or energy and fatigue scores. In line with our scores, they confirmed that baseline physical functions were highest for men who underwent prostatectomy and lowest for those who underwent radiotherapy. Again, in the Protect study, no significant differences among the treatment groups in the physical and mental health sub-scores of the SF-12 scale were found [38]. Similar results at 2 years were reported by a multicentric Spanish trial using the SF-36 scale [39].

Considering factors associated with worse trajectories, a systematic review conducted by Vissers et al. found that cancer patients with diabetes had lower physical functioning and vitality [40]. Another study further demonstrated that cancer patients with diabetes had significantly lower levels of physical function and mental health over time compared to those without diabetes [41], and this result was partially confirmed in PCa patients [42], in accordance with our results. Reeve and colleagues evaluated longitudinally the impact of comorbidities evaluated with the Index of Coexistent Diseases on QoL, and they found a significant impact on physical component but no effect on the mental status [43], in contrast with our findings which was supported, instead, by Chambers et al. [4].

### 4.1. Comparison with ISTAT Study

Finally, we compared our findings with those of the ISTAT's survey focusing on the health state of the Italian population. For patients under 65, there was considerable overlap between the MCS distributions in the ISTAT (in the patients without cancer) and Pros-IT population. As far as the physical domain was concerned, the overlap in distribution was restricted to the group of patients under 55; for the other groups, the physical health condition of the current Italian prostate cancer patient was better than that of the average population 20 years ago. Concerning cancer (all types) patients investigated by the ISTAT study, the patients in Trajectory 1 showed better physical and mental scores. Given its

favourable prognosis, the low impact of treatments' side effects (compared to others), the lack of chemotherapy, and a longer life expectancy, non-metastatic PCa patients seems to have a more negligible effect on mental and physical health than other malignancies [17]. This distance is much more evident if we compare patients with the absence of any other comorbidities. Indeed, both the analysed cohorts have highlighted how the presence of two or more comorbidities can reduce the mental and physical score by 20 to 30 points (see also Table in Supplementary Materials). Although the ISTAT analysis is out of date, recent studies [38,44,45] proved that physical and mental components of health in low-intermediate risk PCa are very similar to those reported by the general population.

### 4.2. Study Limitations

The current study has some limitations. First, since the participating centres were involved voluntarily, a selection bias cannot be excluded. Second, just as for all observational studies, it may be susceptible to confounders. Third, the information on patients experiencing supportive care during follow-up was not included in the data collection. Moreover, data on patients' income and social networks, that might have affected QoL, were not considered and thus not available for analysis. Fifth, there was considerable variability in the times between the diagnosis and the onset of each treatment type; the models were anyway adjusted considering the temporal distance between the end of treatment and the follow-up assessment. Furthermore, different combinations of RT and ADT in terms of starting time and duration is another unmeasured confounding factor for our study. Finally, even if SF-12 has been proved to be responsive to positive change in patients with improved general health and performed well in distinguishing between patients who had improvement in general health and those with worsened general health, caution should be used to evaluate positive change in SF-12 since they could be too responsive to detect "noise" and not clinically significant differences [46]; however, in our analyses on SF-12 changes over time, we did not consider only statistically significant differences, but also MCID, which represent the differences that should be considered as clinically meaningful.

### 5. Conclusions

The study indicates that the vast majority of PCa patients, excluding those with distant metastasis at diagnosis and those treated with chemotherapy or ADT, appear to find a good psychological state once they have come to terms with the diagnosis and have begun their course of treatment. No clinical differences in mental and physical health were found in the various treatments groups at the time of enrolment. Age, diabetes, number of comorbidities, family history of PCa and bowel/urinary dysfunctions were the patient/clinical factors most influencing the probability of deviating from the high mental and physical health. The trajectory analysis in the two years after the cancer diagnosis highlighted the importance of assessing mental health, coping strategies and psychological and interpersonal resources at PCa diagnosis to identify patients who may benefit from personalised support. At diagnosis, patients with impaired mental health (1 over 4) could take advantage of different level of intervention. For patients with transient distress (trajectory 1 in our study), information leaflets or discussions with peers (other PCa patients) and cancer specialist staff could reduce the baseline gap with patients in stable mental condition. For patients with persistent mild distress, the decision making and the treatment itself could not be sufficient to restore a good mental state and level 2 intervention (mild care) with support from a non-oncology team supervised by a clinical psychologist could be of help to reduce differences at two years from diagnosis. The analysis suggested that following these patients between the diagnosis and the treatment could allow for discriminating between those with a good array of resources (trajectory 2) and those more vulnerable (trajectory 1).

**Supplementary Materials:** The following supporting information can be downloaded at: https://www.mdpi.com/article/10.3390/curroncol29110651/s1, Figure S1: The MCS (m) and PCS (p) scores of the Pros-IT patients with color density distribution; Table S1: Responses to the 12-Item Short Form Survey (SF-12) by the Pros-IT patients at the time of diagnosis (*n* (%)); Table S2: Characteristics at the time of diagnosis of the Pros-IT patients classified according to the three trajectories identified by the Mental Component Score (MCS) of the Short-Form Health Survey (SF-12); Table S3: Characteristics at the time of diagnosis of the Pros-IT patients classified according to the two trajectories identified by the Physical Component Score (PCS) of the Short-Form Health Survey (SF-12); Table S4: Mean and standard deviation by age group for MCS-12 and PCS-12 in Pros-IT and ISTAT studies; Table S5: Quartiles divided by age groups for PCS and MCS SF-12 distributions in Pros-IT and ISTAT studies; Table S6: Mean and standard deviation among oncologic Italian citizens (ISTAT) and among Pros-IT patients in trajectory groups identified for MCS-12 and PCS-12; Table S7: Quartiles for MCS and PCS SF-12 distributions for Pros-IT patients in each trajectory groups identified and for oncologic Italian citizens (ISTAT); Table S8: The number of comorbidities (mean and standard deviation (SD)) in the Pros-IT population according to the PCS and MCS deciles; Table S9: The number of comorbidities (mean and standard deviation (SD)) in the ISTAT population according to the PCS and MCS deciles.

**Author Contributions:** Conceptualization, M.N., A.C. and S.M.; formal analysis, M.N. and A.C.; writing—original draft preparation, M.N., A.C., P.D., L.D.L. and S.M.; writing—review and editing, B.N.C., R.S., F.B., S.M., M.G., S.S., F.S., L.B., R.M., P.B., M.M. and R.V.; supervision, S.M. and R.V. All authors have read and agreed to the published version of the manuscript.

**Funding:** Pros-IT CNR is a non-profit observational study. Takeda Italia S.p.A. provided an unconditional grant to the CNR to cover the costs of the preparatory meetings, the meetings of the advisory committees and the PIs, and the cost of developing a web platform for data entry.

**Institutional Review Board Statement:** The Ethics Committee of the coordinating centre (Sant'Anna Hospital, Como, Italy; register number 45/2014) and all the hospitals or health care facilities involved in the project approved the study protocol.

**Informed Consent Statement:** Informed consent was obtained from all subjects involved in the study.

**Data Availability Statement:** The datasets generated and/or analysed during the current study are not publicly available due to the study protocol, which planned that data would be available only to the collaborating scientists within the study.

**Acknowledgments:** The authors wish to thank Linda Moretti Inverso for reviewing the English version of this paper. The Pros-IT CNR Group: Anna Rita Alitto (Roma); Enrica Ambrosi (Brescia); Alessandro Antonelli (Brescia); Cynthia Aristei (Perugia); Michele Barbieri (Napoli); Franco Bardari (Asti); Lilia Bardoscia (Reggio Emilia); Salvina Barra (Genova); Sara Bartoncini (Torino); Umberto Basso (Padova); Carlotta Becherini (Firenze); Rita Bellavita (Perugia); Franco Bergamaschi (Reggio Emilia); Stefania Berlingheri (Brescia); Alfredo Berruti (Brescia); Barbara Bigazzi (Firenze); Marco Borghesi (Bologna); Roberto Bortolus (Pordenone); Valentina Borzillo (Napoli); Davide Bosetti (Milano); Giuseppe Bove (Foggia); Pierluigi Bove (Roma); Maurizio Brausi (Modena); Alessio Bruni (Modena); Giorgio Bruno (Ravenna); Eugenio Brunocilla (Bologna); Alberto Buffoli (Brescia); Michela Buglione (Brescia); Consuelo Buttigliero (Torino); Giovanni Cacciamani (Verona); Michela Caldiroli (Varese); Giuseppe Cardo (Bari); Giorgio Carmignani (Genova); Giuseppe Carrieri (Foggia); Emanuele Castelli (Torino); Elisabetta Castrezzati (Brescia); Gianpiero Catalano (Milano); Susanna Cattarino (Roma); Francesco Catucci (Roma); Dario Cavallini Francolini (Pavia); Ofelia Ceccarini (Bergamo); Antonio Celia (Vicenza); Francesco Chiancone (Napoli); Tommaso Chini (Firenze); Claudia Cianci (Pisa); Antonio Cisternino (Foggia); Devis Collura (Torino); Franco Corbella (Pavia); Matteo Corinti (Como); Paolo Corsi (Verona); Fiorenza Cortese (Alessandria); Luigi Corti (Padova); Cosimo De Nunzio (Roma); Olga Cristiano (Avellino); Rolando D'Angelillo (Roma); Luigi Da Pozzo (Bergamo); Daniele D'agostino (Padova); David D'Andrea (Bolzano); Matteo Dandrea (Padova); Michele De Angelis (Arezzo); Ottavio De Cobelli (Milano); Bernardino De Concilio (Vicenza); Antonello De Lisa (Cagliari); Stefano De Luca (Torino); Agostina De Stefani (Bergamo); Chiara Lucrezia Deantoni (Milano); Claudio Degli Esposti (Bologna); Anna Destito (Catanzaro); Beatrice Detti (Firenze); Nadia Di Muzio (Milano); Andrea Di Stasio (Alessandria); Calogero Di Stefano (Ravenna); Danilo Di Trapani (Palermo); Giuseppe Difino (Foggia); Marco Fabiano (Napoli); Giuseppe Facondo (Roma); Sara Falivene (Napoli); Giuseppe Farullo (Roma); Paolo Fedelini (Napoli); Ilaria Ferrari (Varese);

Francesco Ferrau (Messina); Matteo Ferro (Milano); Andrei Fodor (Milano); Francesco Fontana (Novara); Francesco Francesca (Pisa); Giulio Francolini (Firenze); Giovanni Frezza (Bologna); Pietro Gabriele (Torino); Maria Galeandro (Reggio Emilia); Elisabetta Garibaldi (Torino); Pietro Giovanni Gennari (Arezzo); Alessandro Gentilucci (Roma); Alessandro Giacobbe (Torino); Laura Giussani (Varese); Giuseppe Giusti (Cagliari); Paolo Gontero (Torino); Alessia Guarneri (Torino); Cesare Guida (Avellino); Alberto Gurioli (Torino); Dorijan Huqi (Bolzano); Ciro Imbimbo (Napoli); Gianluca Ingrosso (Roma); Cinzia Iotti (Reggio Emilia); Corrado Italia (Bergamo); Pierdaniele La Mattina (Milano); Enza Lamanna (Ravenna); Luciana Lastrucci (Arezzo); Grazia Lazzari (Taranto); Fabiola Liberale (Biella); Giovanni Liguori (Trieste); Roberto Lisi (Roma); Frank Lohr (Modena); Riccardo Lombardo (Roma); Jon Lovisolo (Varese); Giuseppe Mario Ludovico (Bari); Nicola Macchione (Novara); Francesca Maggio (Imperia); Michele Malizia (Bologna); Gianluca Manasse (Perugia); Giovanni Mandoliti (Rovigo); Giovanna Mantini (Roma); Luigi Marafioti (Cosenza); Luisa Marciello (Prato); Alberto Mario Marconi (Varese); Antonietta Martillotta (Cosenza); Salvino Marzano (Prato); Stefano Masciullo (Bergamo); Gloria Maso (Verbania); Adele Massenzo (Cosenza); Ercole Mazzeo (Modena); Luigi Mearini (Perugia); Serena Medoro (Ferrara); Rosa Molè (Catanzaro); Giorgio Monesi (Novara); Emanuele Montanari (Milano); Franco Montefiore (Alessandria); Giampaolo Montesi (Rovigo); Giuseppe Morgia (Catania); Gregorio Moro (Biella); Giorgio Muscas (Cagliari); Daniela Musio (Roma); Paolo Muto (Napoli); Giovanni Muzzonigro (Ancona); Giorgio Napodano (Salerno); Carlo Luigi Augusto Negro (Asti); Mattia Nidini (Mantova); Maria Ntreta (Bologna); Marco Orsatti (Imperia); Carmela Palazzolo (Messina); Isabella Palumbo (Perugia); Alessandro Parisi (Bologna); Paolo Parma (Mantova); Nicola Pavan (Trieste); Martina Pericolini (Roma); Francesco Pinto (Roma); Antonio Pistone (Salerno); Valerio Pizzuti (Grosseto); Angelo Platania (Messina); Caterina Polli (Prato); Giorgio Pomara (Pisa); Elisabetta Ponti (Roma); Antonio Benito Porcaro (Verona); Francesco Porpiglia (Torino); Dario Pugliese (Roma); Armin Pycha (Bolzano); Giuseppe Raguso (Reggio Emilia); Andrea Rampini (Arezzo); Donato Franco Randone (Torino); Valentina Roboldi (Bergamo); Marco Roscigno (Bergamo); Maria Paola Ruggieri (Reggio Emilia); Giuseppe Ruoppo (Reggio Emilia); Roberto Sanseverino (Salerno); Anna Santacaterina (Messina); Michele Santarsieri (Pisa); Riccardo Santoni (Roma); Giorgio Vittorio Scagliotti (Torino); Mauro Scanzi (Brescia); Marcello Scarcia (Bari); Riccardo Schiavina (Bologna); Alessandro Sciarra (Roma); Carmine Sciorio (Lecco); Tindaro Scolaro (La Spezia); Salvatore Scuzzarella (Lecco); Oscar Selvaggio (Foggia); Armando Serao (Alessandria); Sergio Serni (Firenze); Marco Andrea Signor (Udine); Mauro Silvani (Biella); Giovanni Silvano (Taranto); Franco Silvestris (Bari); Claudio Simeone (Brescia); Valeria Simone (Bari); Girolamo Spagnoletti (Foggia); Matteo Giulio Spinelli (Milano); Luigi Squillace (Pavia); Vincenzo Tombolini (Roma); Mariastella Toninelli (Brescia); Luca Triggiani (Brescia); Alberto Trinchieri (Lecco); Luca Eolo Trodella (Roma); Lucio Trodella (Roma); Carlo Trombetta (Trieste); Marcello Tucci (Torino); Daniele Urzì (Catania); Riccardo Valdagni (Milano); Maurizio Valeriani (Roma); Maurizio Vanoli (La Spezia); Elisabetta Vitali (Bergamo); Stefano Zaramella (Novara); Guglielmo Zeccolini (Vicenza); Giampaolo Zini (Ferrara).

**Conflicts of Interest:** The authors declare no conflict of interest.

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
