# Peer review of "Patient-Factors Influencing the 2-Year Trajectory of Mental and Physical Health in Prostate Cancer Patients"

_curroncol, doi:10.3390/curroncol29110651_

Round 1

Reviewer 1 Report

Thank you for the opportunity to review the manuscript entitled, " Patient-factors influencing the 2-year trajectory of mental and physical health in prostate cancer patients" The authors present an analysis of 1158 men treated for prostate cancer (prostatectomy, radiotherapy, radiotherapy, and androgen deprivation, or active surveillance) and the physical and mental quality of life trajectories monitored over two years. Data were from the Pros-IT CNR trial. The authors found that two years after diagnosis, the psychological and physical status was moderately compromised in more than 10% of the patients. The clinical topic is essential, as the mental well-being of patients diagnosed with cancer is important. However, I have several comments to improve the quality of the manuscript.

  1. It would strongly benefit the paper if the authors added the following to the abstract. First, could the authors present quantitative results? Second, could the authors add how many patients were included in the study? Third, it is slightly confusing from reading the abstract alone if patients were treated or not when the quality of life was assessed. 
  2. Could the authors add quantitative numbers for lines 38-39 when talking about the incidence of prostate cancer?
  3. It will benefit the paper if the authors acknowledge in the introduction that early diagnosis does not necessarily result in increased survival in patients with prostate cancer. 
  4. Could the authors expand on the second paragraph of the introduction? I am missing a discussion in the introduction about prior work, its pros and cons, and what this study adds to the field that the previous ones did not.
  5. Bias related to surveys? Are there any biases associated with the reported quality of life scores? Would it benefit the paper if the authors included potential bias regarding how the scores are grouped in SF-12?
  6. How much data was missing? Why did the authors not impute missing data?
  7. For figure 1, could the authors add what the inclusion criteria were?
  8. Many more people were available after radiotherapy compared to active surveillance. Could there be differential bias?
  9. Did any patients present with metastatic disease at presentation?

Author Response

Reviewer 1:

Thank you for the opportunity to review the manuscript entitled, " Patient-factors influencing the 2-year trajectory of mental and physical health in prostate cancer patients" The authors present an analysis of 1158 men treated for prostate cancer (prostatectomy, radiotherapy, radiotherapy, and androgen deprivation, or active surveillance) and the physical and mental quality of life trajectories monitored over two years. Data were from the Pros-IT CNR trial. The authors found that two years after diagnosis, the psychological and physical status was moderately compromised in more than 10% of the patients. The clinical topic is essential, as the mental well-being of patients diagnosed with cancer is important. However, I have several comments to improve the quality of the manuscript.

Reply: We thank the Reviewer for the comments and welcome the opportunity to address feedback received to strengthen the manuscript.

1. It would strongly benefit the paper if the authors added the following to the abstract. First, could the authors present quantitative results? Second, could the authors add how many patients were included in the study? Third, it is slightly confusing from reading the abstract alone if patients were treated or not when the quality of life was assessed.

Reply: We thank the Reviewer for these suggestions. We have now added in the abstract more details related to the results (percentages of patients in each trajectory) and the number of patients considered in the analyses. We have also specified that quality of life was evaluated at the diagnosis (naïve patients) and two years later.

2. Could the authors add quantitative numbers for lines 38-39 when talking about the incidence of prostate cancer?

Reply: We have now added, as suggested, data related to the global incidence of prostate cancer (page 1 “Prostate cancer (PCa) is one of the leading cancers diagnosed in adult males living in developed countries worldwide, with an estimated incidence of 1,436,000 new cases and an age-standardised incidence rate of 49.9 per 100,000 person-years”).

3. It will benefit the paper if the authors acknowledge in the Introduction that early diagnosis does not necessarily result in increased survival in patients with prostate cancer.

Reply: As suggested, we added a sentence in the Introduction (pages 1-2 “Even if the stage of cancer detection with an early diagnosis is important for most cancer survival, for PCa, considering the extremely high one and five year survival rates, the stage of detection could be less important [Hawkes N. Cancer survival data emphasise importance of early diagnosis. BMJ. 2019 Jan 25;364:l408. doi: 10.1136/bmj.l408]”).

4. Could the authors expand on the second paragraph of the Introduction? I am missing a discussion in the Introduction about prior work, its pros and cons, and what this study adds to the field that the previous ones did not.

Reply: We thank the Reviewer for this suggestion. The Introduction has been updated.

5. Bias related to surveys? Are there any biases associated with the reported quality of life scores? Would it benefit the paper if the authors included potential bias regarding how the scores are grouped in SF-12?

Reply: As demonstrated by Choi EP et al. (2016), SF-12 has been proved to be responsive to positive change in patients with improved general health, and performed well in distinguishing between patients who had improvement in general health and those with worsened general health. However, caution should be used to evaluate positive change in SF-12 since they it could be too responsive to detect “noise” and not clinically significant differences. For this reason, in our analyses on SF-12 changes over time we did not considered only statistically significant differences, but also minimally clinically important differences (MCID), which represent the differences that should be considered as clinically meaningful. A sentence has been added in the Discussion to describe this aspect among limitations of the study (page 14).

6. How much data was missing? Why did the authors not impute missing data?

Reply: As described in Figure 1, no missing data were found at the 6-month follow-up, while 11% and 31% of missing data were found for the 12- and the 24-month assessments, respectively. We preferred not to impute missing data since, as described in Arrandale V et al. (Arrandale V, Koehoorn M, MacNab Y, Kennedy SM (2006). How to use SAS Proc Traj and SAS Proc Glimmix in respiratory epidemiology. doi:http://dx.doi.org/10.14288/1.0048205), the procedure considered to define trajectories (Proc Traj) is able to handle missing data, considering the maximum likelihood method to estimate parameters, evaluating group sizes and shapes of the trajectories.

7. For figure 1, could the authors add what the inclusion criteria were?

Reply: Figure 1 has been modify as suggested.

8. Many more people were available after radiotherapy compared to active surveillance. Could there be differential bias?

Reply: Follow-up participation is different across groups defined by PCa treatments; however, as described in Table 1 and also in our prevous manuscript (Palumbo C, Bruni A, Antonelli A, et al. Health-related quality of life 24 months after prostate cancer diagnosis: an update from the Pros-IT CNR prospective observational study. Minerva Urol Nephrol. 2022 Feb;74(1):11-20), participants’ characteristics were already different at diagnosis, and those treated with radiotherapy were older, had higher comorbidities and higher-risk disease features with respect to those treated with radical prostatectomy (nerve sparing or non-sparing) or in active surveillance. This point has been added (page 5).

9. Did any patients present with metastatic disease at presentation?

Reply: We thank the Reviewer for the opportunity to clarify this aspect. 32 patients had distant metastasis at diagnosis and were not included in the analyses (as can be seen in the Figure 1); a sentence has been added also in the manuscript to clarify this point (page 4).

Reviewer 2 Report

Diagnosis of cancer is very traumatic for patients and prostate cancer (PCa) is no exception. Due to modern lifestyle and fast paced life, hundreds of thousands of new PCa patients have been diagnosed every year. Several questions come to the mind of these patients related to their quality of life and treatment related difficulties which leads to mental and physical distress. In this article, authors analyzed the mental and physical well being of PCa patients over the two years period from the diagnosis of their disease. Although the findings are not surprising still the findings are relevant. Authors conclude that help from clinical psychologists will be good for these patients and there is no doubt that a better mental state will help them overcome the downgrade in quality of life especially after commencement of ADT. Although authors identified some limitations in their study especially the information on patients with supportive care which may help these patients with a better mental state and quality of life; still this article should be accepted for publication in its current form so that this analysis could be made available to clinicians worldwide.   

Author Response

Diagnosis of cancer is very traumatic for patients and prostate cancer (PCa) is no exception. Due to modern lifestyle and fast paced life, hundreds of thousands of new PCa patients have been diagnosed every year. Several questions come to the mind of these patients related to their quality of life and treatment related difficulties which leads to mental and physical distress. In this article, authors analysed the mental and physical well being of PCa patients over the two years period from the diagnosis of their disease. Although the findings are not surprising still the findings are relevant. Authors conclude that help from clinical psychologists will be good for these patients and there is no doubt that a better mental state will help them overcome the downgrade in quality of life especially after commencement of ADT. Although authors identified some limitations in their study especially the information on patients with supportive care which may help these patients with a better mental state and quality of life; still this article should be accepted for publication in its current form so that this analysis could be made available to clinicians worldwide.

Reply:

We thank the Reviewer for the positive appraisal of the manuscript.

Reviewer 3 Report

This paper has some broad problems that limit its value to the oncological community. I’ve highlighted in italics some points that need to be addressed.

For example…

In the first paragraph, the authors cite three papers as background to the thesis that prostate cancer (PCa) and it’s treatment distress PCa patients. One of those references is from Asia. Another is from Australia. Only one reference is recent. In the same paragraph, the author say that “depression and anxiety appear to be their highest levels during the initial treatment phase [of PCa].” This needs to be referenced in light of the more recent literature. It is too much of a sweeping generality and likely not true for patients with symptomatic metastatic disease. Overall, the literature that is cited is an insufficient representation of the literature on the topic of PCa and psychological distress. The ms. needs a much more extensive and up to date literature review.

I am not statistically savvy and thus not qualified to assess the data from the growth mixture models and the trajectories that are the core results of this study.  But I suspect that some of the authors, particularly those that wrote the Discussion, are not that familiar with it either (see below). To get educated—and to find out how common this approach is within oncological research—I put the following string of words into PubMed:

"growth mixture models" cancer trajectories

I got only 21 references and most of them were not relevant to understanding the methodology in this paper. As such, I suggest that the authors provide a more extensive explanation for their statistical methodology keeping in mind the focus of Current Oncology. That additional text should be in easily assessable language that a non-statistically savvy, clinical oncologist might understand.

Although the paper ends by promoting the idea that “a non-oncologist team supervised by a clinical psychologist could be of help [to PCa patients]”, this topic is not mentioned at all in the Introduction. In addition, when it shows up in the concluding paragraph of the Conclusions, it is not supported by any results of the paper. Either data supporting it should be added to the paper and introduced in the Introduction or the idea should be cleaned out of the paper.

Many of the result are obvious and already well-established in the literature. So, for example, patients who have had a nerve-sparing radical prostatectomy that was effective, retain penile erections, As such, they do psychologically better than those who have a permanent erectile dysfunction. We also know—and have for a long time—that urinary incontinence is emotionally brutal for PCa patients. Similarly, it is hardly a novel finding that patients, who are diabetic, have poor health status when diagnosed and treated for PCa with treatments that deprive them of their erections, do worse than patients that are not diabetic and have a nerve-sparing radical prostatectomy.

Against this background the authors extensively massage their data to show that urinary and sexual function are adversely impacted by PCa treatments and reduce the quality of life of PCa patients. This is only worth publishing, if the authors compare their findings with what is already in the literature. Here the discuss needs to be more than just a list of citations.  The authors need to discuss how their results are similar or different from those in other studies with other PCa patient populations,.

In general the authors make no effort to explore any areas where their findings different from those of other studies in the literature. The paper needs a much more extensive literature review at the front end and a discussion that compares their findings to other findings that are in the literature.

The authors set themselves up for this criticism when they flag a study from Australia and one from Asia to claim at the outset the PCa treatments impact patients’ quality of life. That begs for them to compare how similar and different the Italian population is from the Asian, Australian, and other PCa populations.

Another problem with the paper is the way the authors present date on PCa patients receiving ADT. It is already known that ADT has a negative impact on the psychological status of PCa patients. Perhaps that is why the authors exclude from their study patients who are on ADT as a single treatment. However, their data on patients receiving ADT with radiotherapy approaches significance (P = 0.1). The reader does not know what to make of those data because we do not know how long those patients were on ADT. We also don’t know the time period when they were on that treatment in relation to the times when their physical and psychological performance was assessed. Perhaps the authors should leave out all ADT related data.

The third paragraph of the Discussion further emphasizes the inadequate nature of the literature review in this paper. They we are told, for example, that the patients “were meeting the cancer head–on and taking concrete steps to protect their health.“ That is a subjective assessment. I saw no data in the ms from which that conclusion could be justifiably extracted.  Indeed, the authors provide no data on patient’s confidence in decision-making nor self-efficacy and acceptance of health outcomes from treatment. Without a review of the relevant literature, the authors state on the bottom of page 11 that their data “have confirmed that psychological decision-related distress in men diagnosed with PCa diminishes with time, independent of the treatment undertaken”. That claim is supported by a single reference from 15 years ago. More recent studies have come up with quite different conclusions in line with the treatment side effects burden on the patients. None of that literature is discussed, but all of it should be!

In the fifth paragraph of Discussion, the authors see a “need to identify the best coping strategies for each patient and learn how men diagnose it PCa psychologically adjust to cancer.“ In the last 15 years there’s been many papers discussing this exact issue, but none of the relevant papers are discussed in the ms.

A truly surprising claim is made near the bottom of the second to last paragraph in the Discussion. There are the authors say that “PCa seems to have a more negligible effect on mental and physical health than other malignancies”. This statement needs to be much refined. Are the authors comparing patients at the same stage of the malignancy? Are they comparing PCa only to patients who can be treated with curative intent? How did those data compare with studies other than reference #27, which appears to be limited to an Italian data?

Whereas the results overall are torturously detailed, the discussion goes off the rails with broad sweeping generalities that are misleading when one considers the impact of things like erectile disfunction, urinary incontinence, plus depression and metabolic syndrome from hormonal therapy. It is simply beyond the data in this paper for the authors to conclude that “PCa patients have better mental health with respect to patients with other kinds of cancer”.

In the critical last paragraph of the paper, the authors say that their “study indicates that the vast majority of PCa patients appear to find a good psychological state once they have come to terms with the diagnosis and have begun the course of treatment”. This conclusion is not justified given that authors excluded patients diagnosed with systemic disease, and treated with hormone therapy and chemo. Both their own data and much of a published literature suggest that the patients can be deeply burdened by the side effects of those treatments.

As a minor problem, whoever decided that figure 4 was acceptable at the size that it is in the manuscript should be encouraged to think about readability while parking pictures in manuscripts. If they happen to be an MDPI employee, they should not be working for the company. It’s offensive to readers like me to see figures in mss. presented with such low resolution.

In sum it appears that we have in this ms. massive diffusion of responsibility. It appears that the authors who wrote the Discussion were different from the authors that wrote the Results. It’s not clear that the authors of the two sections bothered to read the text generated by the authors of the other section.

Author Response

RESPONSES TO THE REVIEWERS’ COMMENTS

This paper has some broad problems that limit its value to the oncological community. I’ve highlighted in italics some points that need to be addressed.

Reply: We thank the Reviewer for the constructive comments.

For example…In the first paragraph, the authors cite three papers as background to the thesis that prostate cancer (PCa) and it’s treatment distress PCa patients. One of those references is from Asia. Another is from Australia. Only one reference is recent. In the same paragraph, the author say that “depression and anxiety appear to be their highest levels during the initial treatment phase [of PCa].” This needs to be referenced in light of the more recent literature. It is too much of a sweeping generality and likely not true for patients with symptomatic metastatic disease. Overall, the literature that is cited is an insufficient representation of the literature on the topic of PCa and psychological distress. The ms. needs a much more extensive and up to date literature review.

Reply: As also suggested by Reviewer 1, we have updated this section.

I am not statistically savvy and thus not qualified to assess the data from the growth mixture models and the trajectories that are the core results of this study. But I suspect that some of the authors, particularly those that wrote the Discussion, are not that familiar with it either (see below). To get educated—and to find out how common this approach is within oncological research—I put the following string of words into PubMed: "growth mixture models" cancer trajectories. I got only 21 references and most of them were not relevant to understanding the methodology in this paper. As such, I suggest that the authors provide a more extensive explanation for their statistical methodology keeping in mind the focus of Current Oncology. That additional text should be in easily assessable language that a non-statistically savvy, clinical oncologist might understand.

Reply: We thank the Reviewer for this suggestion. The paragraph on trajectories analysis has been better detailed (page 4).

Although the paper ends by promoting the idea that “a non-oncologist team supervised by a clinical psychologist could be of help [to PCa patients]”, this topic is not mentioned at all in the Introduction. In addition, when it shows up in the concluding paragraph of the Conclusions, it is not supported by any results of the paper. Either data supporting it should be added to the paper and introduced in the Introduction or the idea should be cleaned out of the paper.

Reply: We have now updated both the Introduction and the Conclusion.

The results supporting the inclusion of the sentence are related to the differences in the longitudinal trend for trajectories 1 and 2 for MCS; following these patients during the first six months from the diagnosis could allow differentiating between those with a good array of resources (trajectory 2) and those more vulnerable (trajectory 1).

Many of the result are obvious and already well-established in the literature. So, for example, patients who have had a nerve-sparing radical prostatectomy that was effective, retain penile erections, As such, they do psychologically better than those who have a permanent erectile dysfunction. We also know—and have for a long time—that urinary incontinence is emotionally brutal for PCa patients. Similarly, it is hardly a novel finding that patients, who are diabetic, have poor health status when diagnosed and treated for PCa with treatments that deprive them of their erections, do worse than patients that are not diabetic and have a nerve-sparing radical prostatectomy.

Against this background the authors extensively massage their data to show that urinary and sexual function are adversely impacted by PCa treatments and reduce the quality of life of PCa patients. This is only worth publishing, if the authors compare their findings with what is already in the literature. Here the discuss needs to be more than just a list of citations. The authors need to discuss how their results are similar or different from those in other studies with other PCa patient populations.

In general the authors make no effort to explore any areas where their findings different from those of other studies in the literature. The paper needs a much more extensive literature review at the front end and a discussion that compares their findings to other findings that are in the literature.

Reply: We thank the Reviewer for the suggestion. We have now extended the Discussion, including the comparison with other studies.

The authors set themselves up for this criticism when they flag a study from Australia and one from Asia to claim at the outset the PCa treatments impact patients’ quality of life. That begs for them to compare how similar and different the Italian population is from the Asian, Australian, and other PCa populations.

Reply: We thank the Reviewer for the opportunity to clarify this aspect. The paper by De Sousa et al. is a review of studies published between 1999 and 2011, evaluating the various psychological problems seen in patients with prostate cancer, discussing also the relationship between mental health professionals and urologists in a prostate cancer unit and the multidisciplinary management of this type of cancer.

The paper by Korfage and colleagues is a qualitative interview study exploring the relationship between generic quality of life and specific quality of life aspect (urinary, bowel and sexual function). We believe such studies can be mentioned in the Introduction to give an idea of the problem but a direct comparison with our study is out of the purpose of the paper. The Asian paper by Sim and Cheng was erroneously included in this paragraph, and has now been deleted. The whole Introduction paragraph has been updated.

Another problem with the paper is the way the authors present date on PCa patients receiving ADT. It is already known that ADT has a negative impact on the psychological status of PCa patients. Perhaps that is why the authors exclude from their study patients who are on ADT as a single treatment. However, their data on patients receiving ADT with radiotherapy approaches significance (P = 0.1). The reader does not know what to make of those data because we do not know how long those patients were on ADT. We also don’t know the time period when they were on that treatment in relation to the times when their physical and psychological performance was assessed. Perhaps the authors should leave out all ADT related data.

Reply: We recognize the limitation related to the fact that we do not know the duration of the ADT treatment, nor if it was already concluded during the follow-up assessments. We have now stressed among the limitations of the study that "Different combinations of RT and ADT in terms of starting time and duration is another important unmeasured confounding factor for our study."

The third paragraph of the Discussion further emphasises the inadequate nature of the literature review in this paper. They we are told, for example, that the patients “were meeting the cancer head–on and taking concrete steps to protect their health.“ That is a subjective assessment. I saw no data in the ms from which that Conclusion could be justifiably extracted.  Indeed, the authors provide no data on patient’s confidence in decision-making nor self-efficacy and acceptance of health outcomes from treatment.

Without a review of the relevant literature, the authors state on the bottom of page 11 that their data “have confirmed that psychological decision-related distress in men diagnosed with PCa diminishes with time, independent of the treatment undertaken”. That claim is supported by a single reference from 15 years ago. More recent studies have come up with quite different conclusions in line with the treatment side effects burden on the patients. None of that literature is discussed, but all of it should be!

Reply: We have now reviewed this part of the Discussion (page 12).

In the fifth paragraph of Discussion, the authors see a “need to identify the best coping strategies for each patient and learn how men diagnose it PCa psychologically adjust to cancer.“In the last 15 years there’s been many papers discussing this exact issue, but none of the relevant papers are discussed in the ms.

Reply: We thank the reviewer for this comment. Since coping strategies are beyond the aims of the manuscript, we preferred to delete this sentence from the manuscript.

A truly surprising claim is made near the bottom of the second to last paragraph in the Discussion. There are the authors say that “PCa seems to have a more negligible effect on mental and physical health than other malignancies”. This statement needs to be much refined. Are the authors comparing patients at the same stage of the malignancy? Are they comparing PCa only to patients who can be treated with curative intent? How did those data compare with studies other than reference #27, which appears to be limited to an Italian data?

Reply: More details has been introduced in the ISTAT section of the manuscript. The sentence “PCa seems to have a more negligible effect on mental and physical health than other malignancies” was based on the data reported in the ISTAT study. We have now included more updated references showing that physical and mental components of health in low-intermediate risk PCa patients are very similar to those reported by the general population (page 14).

Whereas the results overall are torturously detailed, the discussion goes off the rails with broad sweeping generalities that are misleading when one considers the impact of things like erectile disfunction, urinary incontinence, plus depression and metabolic syndrome from hormonal therapy. It is simply beyond the data in this paper for the authors to conclude that “PCa patients have better mental health with respect to patients with other kinds of cancer”.

Reply: The sentence refers to the data found in the ISTAT population. We acknowledge that that analysis is not up to date and could not be representative of the current clinical situation. Consequently, we concluded with a comparison with the general population, which was confirmed more recently (page 14).

In the critical last paragraph of the paper, the authors say that their “study indicates that the vast majority of PCa patients appear to find a good psychological state once they have come to terms with the diagnosis and have begun the course of treatment”. This conclusion is not justified given that authors excluded patients diagnosed with systemic disease, and treated with hormone therapy and chemo. Both their own data and much of a published literature suggest that the patients can be deeply burdened by the side effects of those treatments.

Reply: We have now specified that our conclusion refers to the included line of treatments, not considering patients with distant metastasis at diagnosis and those treated with chemotherapy or ADT (page 14).

As a minor problem, whoever decided that figure 4 was acceptable at the size that it is in the manuscript should be encouraged to think about readability while parking pictures in manuscripts. If they happen to be an MDPI employee, they should not be working for the company. It’s offensive to readers like me to see figures in mss. presented with such low resolution.
Reply: We apologise to the reviewer. We have now changed the font size in the original figure.

In sum it appears that we have in this ms. massive diffusion of responsibility. It appears that the authors who wrote the Discussion were different from the authors that wrote the Results. It’s not clear that the authors of the two sections bothered to read the text generated by the authors of the other section.

Reply: We regret that the previous version of the manuscript was difficult to read and non-homogenous. The entire manuscript has been now revised based on reviewers' suggestions.

Round 2

Reviewer 1 Report

I believe the manuscript has been sufficiently improved to warrant publication. All my comments and concerns have been adequately addressed.